# Sweet Aging: Glycocalyx and Galectins in CNS Aging and Neurodegenerative Disorders

**DOI:** 10.3390/ijms26104699

**Published:** 2025-05-14

**Authors:** Mohd Yaqub Mir, Adam Legradi

**Affiliations:** 1Epilepsy Center, Department of Clinical Science, Faculty of Medicine, Lund University, 22184 Lund, Sweden; mohd_yaqub.mir@med.lu.se; 2Department of Cell Biology and Molecular Medicine, University of Szeged, H-6720 Szeged, Hungary

**Keywords:** aging, galectins, neuroinflammation, glycan–lectin interaction

## Abstract

Aging and aging-related neurodegenerative disorders, such as Alzheimer’s disease, are characterized by chronic inflammation that progressively damages nervous tissue within the central nervous system (CNS). In addition to cytokines, lectin-like carbohydrate recognition molecules play a critical role in modifying cellular communication during inflammation. Among these, galectins—particularly anti-inflammatory galectin-1 and pro-inflammatory galectin-3—stand out due to their immunological functions and specificity for N-acetyllactosamine structures. Almost every cell type within the CNS can express and recognize galectins, influencing various essential cellular functions. N-acetyllactosamines, the ligand structures recognized by galectins, are found beneath sialylated termini in protein-linked oligosaccharides. During aging, protein-linked oligosaccharide structures become shorter, exposing N-acetyllactosamines on protein surfaces, which enhances their availability as binding sites for galectins. Genomic studies indicate that the number of galectin-1- and galectin-3-expressing microglial cells increases with age- or age-related disease (Alzheimer’s disease), reflecting an aging-associated rise in galectin concentrations within the CNS. This increase parallels a rise in free N-acetyllactosamine-like ligands, suggesting that galectin-N-acetyllactosamine interactions gain prominence and play a more significant role in aging-related CNS disorders. Understanding these interactions and their molecular implications offers potential avenues for targeted therapeutic strategies in combating aging-related CNS inflammation and neurodegeneration.

## 1. Introduction

Aging is a complex, ever-evolving biological process that profoundly influences our cellular and neurological systems. One hallmark of aging is a subtle but persistent low-grade inflammation termed “inflammaging” [1,2,3,4]. This sterile inflammation, occurring without infection, is a key feature of the aging process and has been implicated in age-related diseases such as Alzheimer’s and Parkinson’s disease, both marked by chronic, low-level central nervous system (CNS) inflammation [3,4,5].

During the aging process, levels of pro-inflammatory cytokines such as IL-1 and IL-6 increase, contributing to neuroinflammation. Evidence also suggests that certain anti-inflammatory cytokines, including transforming growth factor (TGF) and IL-1RA (interleukin-1 receptor antagonist), show elevated levels in aging [6,7,8]. These changes in cytokine levels further shape the inflammatory milieu associated with aging and its impact on the CNS.

Carbohydrate-binding proteins, such as galectins, play significant roles in regulating these inflammatory events. Galectin-1 demonstrates strong anti-inflammatory properties by inducing apoptosis in activated T lymphocytes [9], while galectin-3 exhibits pro-inflammatory functions by inducing the expression of inflammatory cytokines in septic and other serious inflammatory conditions [10]. This dual functionality of galectins underscores their complex involvement in the inflammatory processes of aging.

By examining these intricate mechanisms, we gain insight into how inflammation influences the aging process and age-associated diseases. Such understanding may inform targeted strategies for mitigating inflammation-driven aging effects and their neurological consequences.

## 2. Structure of Glycocalyx and Carbohydrate Recognition Domain Containing (CRD) Proteins

The ligands of these lectins are oligosaccharides attached to the extracellular domains of membrane lipids and proteins. The lipid and protein components of the cell membrane are glycosylated, and these glycan structures on the cell surface form the “glycocalyx” or “cell coat.” The biological functions of glycans can be categorized into three broad areas as follows: (1) structural glycans, which are part of extracellular scaffolds such as cell walls or the extracellular matrix; (2) glycans which participate in energy metabolism as a carbon source for energy production; and (3) glycans with information carriers based on interactions with GBPs (glycan binding proteins) [11].

Glycosylation in the endoplasmic reticulum (ER) and Golgi apparatus produce various glycans that typically attach to cellular proteins and lipids. In mammalian cells, the building blocks of cell surface glycans consist of nine monosaccharides, namely fucose, galactose, N-acetyl-lactosamine, glucose, glucuronic acid, mannose, sialic acid, and xylose. These monosaccharides combine to form the final protein- and lipid-bound glycan structures [12].

In N-glycans, the carbohydrate chain attaches to asparagine residues of proteins, specifically in the Asn-X-Ser/Thr motif. All eukaryotic N-glycans share a common core sequence with five carbohydrate molecules coupled to the Asn amino acid side chain of the Asn-X-Ser motif in the following order: Manα1-3(Manα1-6) Manβ1-4GlcNAcβ1-4GlcNAcβ1-Asn-X-Ser/Thr.

This core sequence could be extended to form the final structure of the cell surface N-glycans. There are oligomannose-type glycans, in which only mannose residues extend the core and complex N-glycans where antennae initiated by GlcNAc extend the core. The structure of complex N-glycans can form bi-, tri-, or tetra-antennae complexes, depending on the number of branches. These branched N-glycans on the cell surface have a high capacity to bind GBPs, enabling information carrier functions. In hybrid N-glycans, mannose extends the Manα1-6 arm of the core, while one or two GlcNAc-initiated antennae extend the Manα1-3 arm [13] (Figure 1).

In O-glycan structures, a glycan structure attaches to Ser/Thr residues, forming four major core structures. The Core 1 glycans are formed by the extension of O-GalNAc linked to Ser/Thr residues by β1-3Gal. The Core 2 O-glycans are formed by the addition of β1-6GlcNAc to the GalNAc of Core 1. In Core 3 glycans, the GlcNAc couple to –O-GalNAc with a β1-3 link. The Core 4-type O-glycans are formed by branching Core 3 with β1-6GlcNAc. Each core can be extended by a variety of sugar residues, forming linear or branched chains similar to those of N-glycans and glycolipids [14].

## 3. Galectins in the Central Nervous System (CNS)

Galectins, the best characterized immunoregulatory lectins, in particular, bind β-galactose-containing glycoconjugates (e.g., N-acetyl-lactosamine (LacNAc), formed by Gal and GlcNAc) and share structural homology in their carbohydrate recognition domains. These are also referred to as S-type lectins, which require a reducing environment to keep glycan-binding activity. The galectin family consists of 15 distinct members, which are classified into three main structural types as follows: the proto-type galectins, with two identical CRDs (galectin-1, -2, -5, -7, -10, -11, -13, -14, and -15); chimera-type galectins, with one CRD and another distinct domain (galectin-3); and the tandem repeat galectins, which possess two CRDs with different carbohydrate specifities (galectin-4, -6,-8,-9, and -12) [15].

The glycan-binding affinity of galectins is influenced by the glycan structure and modifications of galactose residues, such as fucosylation, sialylation, and sulfation [16]. For instance, the affinity of galectins increases with the branching of complex N-glycans or the presence of a repeated LacNAc motif. However, galectins vary in their binding capacities depending on the glycan modification. Modification at the C6 position of galactose generally reduces galectin-binding affinity. Galectin-1 recognizes alpha2, 3-sialylated and the non-sialylated branched N-glycans but not the alpha2, 6 sialylated N-glycans [17]. In contrast, galectin-3 and galectin-9 can tolerate alpha2, 6 sialylated structures to some extent, and their overall binding strength remains decreased compared to alpha2, 3-sialylated structures [18]. The Core 2 O-glycans with polyLacNac structures can also be recognized by galectin-1, galectin-3, and galectin-9 [19]. Galectins also bind to lactose-containing glycolipids like galectin-4, which binds to sulfatide and SM3 (lactosylceramide sulfate); galectin-1 recognizes GM (manosialo-gangliosid) and GA1 (asialo gangliosids)-type gangliosides, and galectin-8 couples to GD1a (ganglioside N-acetylgalactosaminyl) and GM3 (monosialodihexosylganglioside).

Additionally, galectins, especially galectin-3 and galectin-9, are known to bind sulfated proteoglycans, such as keratan sulfate and chondroitin sulfate [20].

Because of their heteromeric structure and multiple CRDs, many galectins can bridge cell surface molecules. This promotes interactions among cell surface components, facilitates lipid raft formation, and influences signal transduction events [21].

The most characterized member of the galectin (galectin-1, galectin-3, galectin-4, galectin-8, and galectin-9) family is expressed in all cell types of the CNS (Table 1).

### 3.1. Neurons and Galectins

#### 3.1.1. Galectin-1 in Neurons

According to in situ hybridization studies, galectin-1 is expressed in primary sensory neurons of the dorsal root ganglia (DRG), motoneurons in the anterior horn of the spinal cord, and motor nuclei in the brainstem, including the facial motor nucleus, trigeminal motor nucleus, and nucleus ambiguus. Additionally, specific regions of the cerebellum, such as the cerebellar dentate nucleus, along with neurons in the trigeminal mesencephalic nucleus and vestibular nucleus, also express galectin-1. These findings indicate that galectin-1 expression is significantly higher in the DRG and spinal cord than in the brain, and its presence is detectable soon after neuronal differentiation [40].

Galectin-1 interacts with glycosylated structures on the neuronal cell surface, playing a key role in adhesion modulation. For instance, galectin-1 expressed in the olfactory bulb promotes olfactory bulb fasciculation [22]. Although the oxidized form of galectin-1 loses its glycan-binding capacity, it promotes neurite outgrowth through an alternative mechanism, thereby supporting axonal regeneration [41,42]. Both galectin-1 and galectin-3 promote neurite outgrowth and branching by inhibiting the Semaphorin 3A (Sema3A) pathway or binding to neural cell adhesion molecule L1 (NCAML1) and laminin. Sema3A normally has a repellent effect on axonal growth, as it inhibits actin polymerization via PlexinA4 surface receptors. Galectin-1 blocks this interaction by binding to the glycan structure on PlexinA4, facilitating its endocytosis and removing it from the cell surface, thereby promoting axon growth [23].

#### 3.1.2. Galectin-3 in Neurons

Galectin-3 is widely expressed in the brain, with elevated levels in the hypothalamus, particularly in the arcuate nucleus and dorsomedial nucleus. Moderate expression is observed in the supraoptic, paraventricular, and ventromedial nuclei. Galectin-3 is also present in the vascular organ of the lamina terminalis (VOLT), a sensory circumventricular organ (CVO). Outside the hypothalamus, galectin-3 is found in the central nucleus of the inferior colliculus, the cochlear nucleus, the solitary nucleus, and the pontine nucleus, though subcortical nuclei show no immunoreactivity [40].

Galectin-3 interacts with laminin and NCAML1, modulating cell adhesion and promoting neurite outgrowth [24]. Its phosphorylated form (pGal-3) enhances NCAML1 association with Thy-1-rich membrane microdomains and recruits membrane–actin linker proteins (ERM (ezrin radixin moesin) proteins) via interaction with the intracellular domain of L1. This mechanism locally regulates the L1–ERM–actin pathway, contributing to axon branching. Phosphorylation of galectin-3 may act as a molecular switch for axonal responses [25].

#### 3.1.3. Galectin-4 in Neurons

Galectin-4 is expressed in the rat brain during early postnatal development but decreases significantly by days 12–16 and becomes undetectable in adults. Its expression is linked to the myelination process and is localized in the cerebral cortex, hippocampal formation, thalamus, and corpus callosum, where it is found in both neurons and oligodendrocytes [37]. Galectin-4 facilitates communication between neurons and oligodendrocytes, which is essential for proper myelination. It binds to the N-glycan end of NCAML1 molecules, promoting the L1 membrane clustering needed for myelin sheath formation. Galectin-4 also binds sulfatide-containing microtubule-associated carriers, organizing the transport of L1 and other axonal glycoproteins, which is critical for proper axon growth [26].

#### 3.1.4. Galectin-8 in Neurons

Galectin-8 is expressed in the thalamus and choroid plexus and is expressed weakly in the hippocampus and cortex in mice. It is produced by primary cultured hippocampal neurons and has neuroprotective effects against glutamate-induced excitotoxicity, oxidative stress, and amyloid-beta-induced neurotoxicity. Galectin-8 achieves this by activating β1-integrins (α3/β1 and α5/β1 integrins), ERK1/2, and PI3K/AKT signaling pathways, which mediate neuroprotection [27].

### 3.2. Astrocytes and Galectins

#### 3.2.1. Galectin-1 in Astrocytes

Astrocytes predominantly express galectin-1, which induces their differentiation while strongly inhibiting proliferation. Differentiated astrocytes, in turn, produce high levels of brain-derived neurotrophic factor (BDNF), a mechanism that may help prevent neuronal loss after injury [28,43]. Following ischemic injury, galectin-1 expression is upregulated in activated astrocytes around the infarct, where it inhibits astrocyte proliferation, attenuates astrogliosis, and downregulates the expression of nitric oxide synthase and interleukin-1β [44]. However, after spinal cord injury, GFAP-positive reactive astrocytes begin to express galectin-1, which enhances astrocytosis [45].

#### 3.2.2. Galectin-3 in Astrocytes

Galectin-3 is also expressed by activated astrocytes in the ischemic brain, where it plays a role in post-ischemic tissue remodeling by enhancing angiogenesis and neurogenesis [46,47]. During brain injury, galectin-3 activates the Notch signaling pathway, which is essential for the astrocytic response to injury by reactive astrocytes [29].

#### 3.2.3. Galectin-9 in Astrocytes

Under normal conditions, astrocytes do not express galectin-9. However, its expression increases in response to inflammatory signals such as TNF-α, IL-1β, and interferon-γ [48].

### 3.3. Microglia and Galectins

#### 3.3.1. Galectin-1 in Microglia

A specific microglia subpopulation with increased galectin-1 expression has been identified in aging and Alzheimer’s disease (AD) cohorts. This heightened galectin-1 expression is associated with the morphologically active, amoeboid phenotype of microglia [49]. Galectin-1 plays a critical role in reducing microglia-related inflammatory processes by deactivating classically activated microglia and promoting an M2 phenotype shift. This reprogramming is mediated by the direct binding of galectin-1 to Core 2 O-glycans on CD45 [30].

#### 3.3.2. Galectin-3 in Microglia

Activated microglia, but not resting microglia, express abundant galectin-3. Its expression increases in both M1 and M2 microglial phenotypes and rises further in pathological conditions such as ischemia or brain injury. Galectin-3 is also released by activated microglia in neurodegenerative diseases, including Alzheimer’s disease, aging, and amyotrophic lateral sclerosis (ALS) [50].

In Alzheimer’s disease, activated microglia expressing galectin-3 bind to TLR4 (Toll-like receptor 4), MerTK, and TREM2 (triggering receptor expressed on myeloid cells 2), which facilitates amyloid-beta clearance and promotes inflammatory responses associated with the disease [31,32,33]. In Parkinson’s disease, the internalization of alpha-synuclein triggers galectin-3 release by microglia [51].

In Huntington’s disease, microglia-released galectin-3 activates inflammasomes, leading to cytokine release [52]. In multiple sclerosis (MS), galectin-3 expression is associated with demyelination and interacts with TREM2 to regulate inflammatory events [53].

In stroke, galectin-3 binds to IGF1R (insulin-like growth factor receptor 1) and TLR4, mediating cytokine release and microgliosis around the infarct site. Additionally, galectin-3 promotes VEGF (vascular endothelial growth factor) release, supporting angiogenesis [54]. In traumatic brain injury (TBI), galectin-3 interacts with TLR4, driving intense inflammatory responses and cytokine release [55].

Galectin-3 also binds to the MerTK receptor on microglial surfaces, enhancing their phagocytic activity. Integrin-linked kinase (ILK) plays a central role in microglial reprogramming mediated by galectin-3 [34].

#### 3.3.3. Galectin-4 in Microglia

While galectin-4 is not expressed under normal microglial conditions [37], it becomes expressed in activated microglia within the brain and spinal cord in cuprizone-induced demyelination models of multiple sclerosis lesions [56].

#### 3.3.4. Galectin-9 in Microglia

Stimulation with Poly I:C triggers galectin-9 expression in microglia, and astrocyte-derived galectin-9 further enhances TNF expression in microglial cells [35].

### 3.4. Oligodendrocytes and Galectins

During their development, oligodendrocytes undergo a well-characterized maturation process, with each stage associated with a specific molecular expression pattern.

The expression of galectin-1 is associated with A2B5/PDGFRα/O4-positive oligodendrocyte precursors and pre-oligodendroblasts but is absent in O1-positive mature oligodendrocytes [36].

Galectin-3 is predominantly expressed in mature oligodendrocytes. Oligodendrocyte progenitor cells (OPCs) secrete matrix metalloproteinase 2 (MMP2), which cleaves galectin-3 into truncated proteins. These truncated forms lack the ability to form oligomers but have an increased carbohydrate-binding capacity [57,58]. Extracellular galectin-3 promotes oligodendrocyte differentiation and myelination, whereas galectin-1 inhibits these processes, suggesting opposing roles for the two galectins in oligodendrocyte differentiation [36].

Galectin-4 is present in the nucleus of OPCs and accumulates in the nucleus of mature oligodendrocytes [37]. Interestingly, galectin-4 is released by neurons—not oligodendrocytes—and inhibits myelination by binding to pre-myelinating oligodendrocytes, thereby influencing the timing of myelination [37,59].

### 3.5. Neural Stem/Progenitor Cells and Galectins

The brain contains two primary populations of stem and progenitor cells, with one in the subventricular zone (SVZ) of the lateral ventricle and another in the subgranular zone (SGZ) of the hippocampal dentate gyrus.

The SVZ consists of astrocytic neural stem cells (Type B1 cells), astrocytes (Type B2 cells), transit-amplifying cells (Type C cells), and neuroblasts (Type A cells). Both galectin-1 and galectin-3 are expressed in this region.

Galectin-1 is expressed in GFAP-positive Type B1 cells (astrocytic neural stem cells). It plays an important role in the proliferation of adult neural progenitor cells, including SVZ astrocytes. Interaction between β1 integrin and galectin-1 is crucial for the proliferation-promoting effects of galectin-1 in neural progenitor cells of the subependymal zone (SEZ) [38].

Galectin-3 is expressed in astrocytes but not in neuroblasts within the SVZ. Its presence promotes cell migration through the EGFR-mediated (epidermal growth factor receptor) pathway [39]. Galectin-3 also inhibits cell emigration from the SVZ in the cuprizone-induced multiple sclerosis model [60]. In viral-induced multiple sclerosis models, galectin-3 is essential for the neurogenic niche response in the SVZ. Loss of galectin-3 reduces chemokine expression, immune cell migration to the SVZ, and progenitor cell proliferation, particularly in the corpus callosum [61].

After stab wound injury, both galectin-1 and galectin-3 expression increase in reactive astrocytes and neural stem cells, highlighting their role in regeneration [62].

The hippocampal dentate gyrus subgranular layer is the second brain region with neural stem/progenitor cells. Galectin-1 promotes both the basal and kainate-induced proliferation of neural progenitor cells in the dentate gyrus, but the presence of galectin-1 inhibits the neural differentiation of the neuronal progenitor cells [63,64].

## 4. Aging and Glycocalyx Modification in Aging-Coupled Inflammatory Events

More than 100 rare human genetic disorders have been identified that result from deficiencies in glycosylation pathways. Many of these disorders impact the central and/or peripheral nervous system, causing symptoms such as intellectual disabilities, hypotonia, and seizures.

One common disorder affecting N-linked glycosylation is PMM2-CDG (phosphomannose mutase2 congenital disorder), caused by a mutation in the PMM2 enzyme, which is responsible for converting mannose-6-phosphate to mannose-1-phosphate. This deficiency disrupts the dolichol-coupled oligosaccharide chain formation in the ER, impairing the global glycosylation process [65].

During aging, the proportion of modified N-glycans, such as monoantennary (A1), agalactosylated (G0), and oligomannose (OM), increases in the serum of both sexes [66]. In immunological terms, immunoglobulins are the most significant serum molecules. Aging causes global changes in IgG-coupled N-glycan patterns, particularly modifications at the ends of oligosaccharides. These changes include reduced galactose and sialic acid residues on branches (leaving GlcNAc in the terminal position) and increased core fucosylation.

These changes are prominent in inflammation-related autoimmune diseases such as rheumatoid arthritis [67]. Elevated levels of agalactosylated IgG-G0 in aged individuals suggest that altered IgG activity may contribute to age-related inflammation, also known as “inflammaging”. Aging alters not only the galactosylation but sialylation patterns of the IgG, which modulate the interaction between IgG and Fc receptors, thereby affecting their efficacy [68].

Aging leads to a reduced expression of glycosylation factors, not only in peripheral tissues but also in the brain. This reduction affects glycan structures and functions [69]. For example, studies have shown that the aging brain exhibits decreased complexity in glycan profiles, particularly in brain-derived CSF (cerebrospinal fluid) proteins. The altered N-glycosylation patterns of these proteins impair their function and cellular communication, potentially affecting synaptic plasticity and neuronal connectivity—key processes for maintaining cognitive function [70,71].

In the brain and myelin structures, glycosylation patterns differ significantly between embryonic and adult stages, suggesting a role for glycosylation in aging.

The neural cell adhesion molecule (NCAM) is a nervous system-specific molecule that mediates cellular adhesion in neural tissues. During embryonic development, the polysialylated form of the NCAM (PolySia-NCAM) is widely and highly expressed in the nervous system, playing crucial roles in neurogenesis, cell migration, axon/dendrite growth, synaptic reorganization, and myelination [72]. In adults, PolySia-NCAM expression is restricted to neurogenic regions, such as the subventricular zone (SVZ) and subgranular zone (SGZ), but it is absent in non-neurogenic regions like the trigeminal ganglion and brainstem nuclei [73].

### Aging-Coupled Chronic Diseases and Glycosylation

Chronic and aging-associated diseases also have glycosylation-dependent molecular backgrounds. Large-scale proteomic analyses have revealed significant changes in N-glycosylation patterns in Alzheimer’s disease. These changes include the appearance of glycosites that are exclusively glycosylated in the AD brain, possibly due to abnormal protein conformational changes that expose new glycosylation sites. Conversely, some proteins lose N-glycosylation sites due to misrecognition by the glycosylation machinery [74]. Alterations in serum protein N-glycosylation have also been observed in other age-related neurodegenerative disorders, including Parkinson’s disease. These findings support the potential use of glycocalyx-related glycosylation changes as early diagnostic biomarkers for Parkinson’s disease [75].

Gene expression studies show that enzymes required for N-glycosylation are dysregulated in AD, altering protein glycosylation [76]. The two key pathogenic proteins in AD, tau and amyloid-beta, both trigger ER stress, disrupting normal ER functions and glycosylation processes [77].

The endothelial glycocalyx is a specialized structure on the surface of endothelial cells that comprises not only conventional lipid- and protein-bound oligosaccharides but also complex proteoglycans such as hyaluronic acid, heparan sulfate, and chondroitin sulfate. This structure plays a critical role in regulating vascular permeability, maintaining hemodynamic stability, and exerting anticoagulant effects through its net negative charge [78]. The integrity of the endothelial glycocalyx is a key determinant of circulatory efficiency and quality, not only in peripheral tissues but also within the central nervous system (CNS).

During aging, the endothelial glycocalyx undergoes structural thinning and compositional changes, including a reduction in mucin-type O-glycans. These alterations compromise blood–brain barrier (BBB) function, increase tissue vulnerability, and elevate the risk of hemorrhage [79]. Structural modifications of the glycocalyx also affect lectin–sugar interactions. For example, plant-derived lectins such as concanavalin A (ConA) can activate inflammasomes and initiate inflammatory responses within the CNS microvasculature. Increased lectin-binding capacity in aging and Alzheimer’s disease (AD) reflects alterations in the endothelial glycocalyx [80,81]. Structural thinning of the endothelial glycocalyx has also been observed in Parkinson’s disease model mice, highlighting the critical role of blood–brain barrier (BBB) integrity in the pathophysiology of Parkinson’s disease [82]. In Parkinson’s disease, aberrant glycation of the toxic substrate α-synuclein enhances its pathogenicity by promoting both direct genomic damage through nuclear interactions and increased production of reactive oxygen species (ROS), ultimately leading to neuronal cell death [83].

Aging also affects the expression of enzymes involved in glycocalyx formation in immune cells, leading to reduced levels of α2,6-linked sialic acid on T cell surfaces, which contributes to T cell exhaustion [84]. Cell surface sialylation plays a central role in immune regulation within the CNS. For instance, microglial activation is associated with increased cell surface sialidase activity, resulting in desialylation that enhances microglial phagocytic capacity [85]. Activated microglia also secrete galectin-3, which binds to desialylated neuronal surfaces, promoting phagocytosis through the Mer tyrosine kinase (MerTK) pathway [86].

These findings suggest that chronic inflammation associated with aging may not only result from increased immune cell infiltration due to a compromised BBB but also from alterations in the immune cell glycocalyx that affect immune activation and regulation.

## 5. Lectin-Based Effect on CNS Aging

As highlighted earlier, glycosylation pattern alterations—particularly in immune-related pathways—play a significant role in chronic inflammation, including aging-related processes. These changes impact the glycocalyx and subsequently alter the amount and quality of membrane-bound lectins during inflammation within the nervous system (Figure 2).

The correct sialylation of cell surface proteins is a crucial regulator of neuroimmunological homeostasis in the CNS. Microglial cells, the main regulator of CNS immune functions, detect sialic acid residues on neuronal membranes via their Siglec receptors. These interactions provide inhibitory signals to microglia due to the ITIM (immunoreceptor tyrosine-based inhibition motif) in the Siglec receptors. When the sialylation pattern is disrupted due to injury or pathological events, this inhibitory signaling is impaired, leading to microglial activation and inflammation-driven tissue damage [87]. Activated microglia lacking ITIM-mediated inhibitory signaling also release galectin-3, which promotes their activation by binding to TLR4 and TREM2 receptors [86].

Beneath the sialylated ends of glycosylated proteins lie N-acetyl-lactosamine structures, which are recognized by galectins. During aging, serum levels of inflammation-linked galectin-3 increase, underscoring their role in aging-related inflammation [88]. Within the CNS, aging alters lectin-like molecules, with increasing levels of galectin-1-expressing microglial populations in aged rodent brains. These microglia exhibit an activated morphology [49].

In Alzheimer’s disease, high levels of galectin-3 exacerbate amyloid deposits and worsen the pathological outcomes of the disease. In contrast, galectin-3 knockout (KO) mice show reduced amyloid-beta oligomerization, suggesting that galectin-3 contributes to the disease’s progression [89]. A microglia population with galectin-3 expression and inflammatory phenotypes was markedly identified in an AD mouse model brain with single-cell RNA sequencing data [90]. Galectin-3 shows colocalization with TREM2 microglial activator receptors in the AD brain [50].

Severely damaged oligosaccharides often terminate in mannose residues, which are recognized by mannose-binding lectins (MBLs). These MBLs can opsonize damaged cells, triggering complement activation or phagocytosis to clear the injured tissue. In AD, MBLs bind to amyloid-beta, modulating inflammation associated with the disease. Elevated levels of MBL may increase the brain’s inflammatory capacity in AD, contributing to disease progression [91].

Aging-associated alterations in the endothelial glycocalyx, as discussed in the previous section, lead to changes in both the quality and quantity of glycocalyx–lectin interactions, thereby modifying lectin-mediated signaling in endothelial cells. In vitro blood–brain barrier (BBB) models suggest that increased shear stress on the surface of endothelial cells alters the gene expression profiles of both glycocalyx-synthesizing enzymes and lectins that interact with the glycocalyx. For example, in such in vitro BBB models, increased flow has been shown to decrease galectin-3 expression in endothelial cells [92]. Galectin-3 localized on the endothelial cell surface interacts with vascular endothelial growth factor (VEGF) receptors, contributing to the stability and maintenance of the microvasculature [93].

## 6. Concluding Remarks

Aging is a highly intricate process, marked by physiological changes across the body, including within the central nervous system. As we age, the delicate molecular balance that supports cellular homeostasis becomes disrupted, leading to impaired cellular functions. Among the first responders in molecular interactions are the glycan proteins and lipids linked with oligosaccharide structures mediated by lectins, which are carbohydrate-recognizing molecules.

The evidence presented highlights that during aging and chronic inflammation, key aspects of lectin-based signaling pathways undergo significant changes. These include alterations to both the receptor component, such as the modified N-glycan patterns on cell surfaces, and the ligand component, as seen in the altered expression of lectins in the CNS. Notably, the presence of simpler glycans with N-acetyllactosamine ends enhances the efficacy of galectin-mediated signaling. These changes in carbohydrate-based interactions could contribute to the disruption of molecular homeostasis, driving the pathological processes associated with CNS aging.

Detecting lectins or their recognized oligosaccharides in cerebrospinal fluid or CNS tissue could serve as a diagnostic tool to assess the state of CNS aging. Furthermore, modulating the strength and efficiency of carbohydrate–lectin interactions presents a potential therapeutic avenue for addressing CNS aging and related inflammatory conditions.

## Figures and Tables

**Figure 1 ijms-26-04699-f001:**
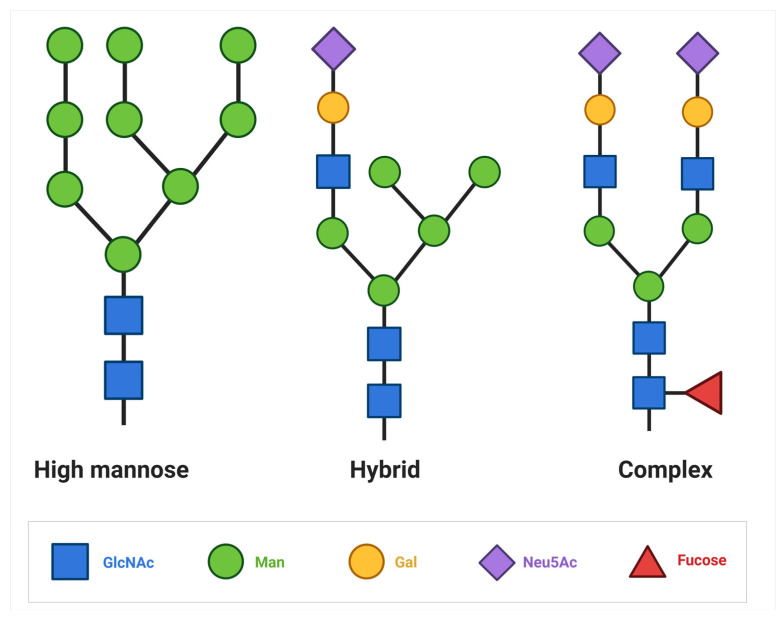
**Type of N-glycans**: In high-mannose N-glycans, the core oligosaccharide sequence extends only mannose residues; in hybrid glycans, one extension contains GlcNAc structures and the other has mannose; in complex glycans, both extensions contain GlcNac structures.

**Figure 2 ijms-26-04699-f002:**
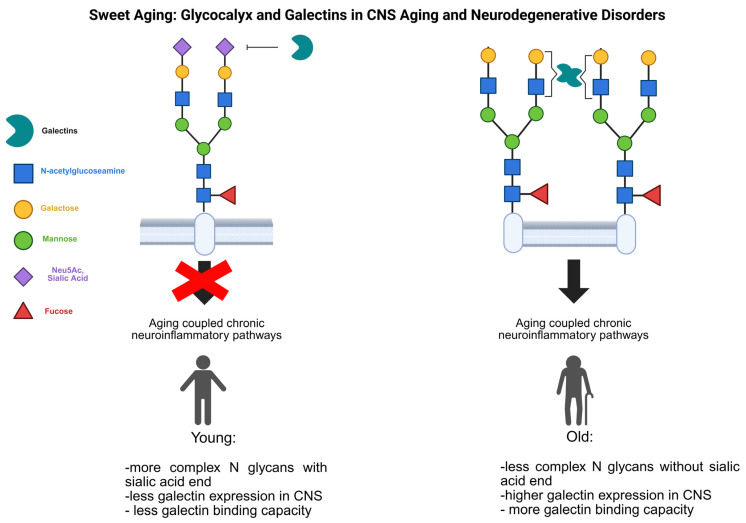
The role of glycan and galectins in aging-coupled neuroinflammatory events. During aging and several age-related disorders, the complexity of protein-linked glycan structures decreased (for example, the terminal sialic acid residue is missing), which opens the way for new lectin–protein interactions (missing terminal in sialic acid increases the galectin-1 binding possibility), and the newly bound lectins reordering the cell surface protein–protein interactions modifying signaling pathways lead to aging-related chronic inflammation. To force this effect, the expression pattern of the lectins was also modified (enhanced expression of galectin-1 by microglial cells).

**Table 1 ijms-26-04699-t001:** The expression pattern of galectins in the CNS.

Cell Type	Galectins	Ligand	Function	References
Neurons	Galectin-1	Lactosamines in glycolipids in olfactory nerve surface	Olfactory axon fascination	[22]
		Plexin A4-coupled glycoconjugates	Promoting injured axon regeneration (neurite outgrowth)	[23]
	Galectin-3	Laminin	Neurite outgrowth in DRG neurons	[24]
		NCAML1 (neural cell adhesion molecule L1)	Axon branching in cultured hippocampal neurons	[25]
	Galectin-4	Sulfatide NCAML1	Axon elongation on hippocampal and cortical neurons	[26]
	Galectin-8	Βα3β1, α5β1 integrins	Neuroprotection in hippocampal neurons	[27]
Astrocytes	Galectin-1	?	Promotes astrocyte differentiation, BDNF secretion, inhibit proliferation	[28]
	Galectin-3	Notch signaling pathway activation	Proper astrocytic answer in ischemic conditions	[29]
Microglia	Galectin-1	CD45-coupled Core 2 O-glycans	M1 microglia deactivation	[30]
	Galectin-3	TLR4, MerTK, TREM2	Promotes amyloid clearance and inflammatory events coupled with Alzheimer’s disease	[31,32,33]
		MerTK	Promotes phagocytosis by ILK pathway	[34]
	Galectin-9	?	Promotes TNF α expression	[35]
Oligodendrocytes	Galectin-1	?	Inhibits oligodendrocyte differentiation and myelination	[36]
	Galectin-3	?	Promotes oligodendrocyte differentiation and myelination	[36]
	Galectin-4	Galactosyl ceramide sulfate	Inhibits myelination	[37]
Neuronal stem and progenitor cells	Galectin-1	Β1 integrin	Regulates the neural stem cell number in SVZ	[38]
	Galectin-3	EGFR	Cell migration events in the SVZ region	[39]

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
