# Peer review of "Sweet Aging: Glycocalyx and Galectins in CNS Aging and Neurodegenerative Disorders"

_ijms, 2025, doi:10.3390/ijms26104699_

Round 1

Reviewer 1 Report

Comments and Suggestions for Authors

In the review article „Sweet Aging: Glycocalyx and Galectins in CNS Aging and Neurodegenerative Disorders“, the authors included the latest knowledge about galectin changes in the CNS with aging.

Below are some points that need attention.

Although the section of the article devoted to the effects of aging on galectins in the CNS is extensive, the section of this article devoted to the effects of aging on the glycocalyx is scarce and should be expanded. I suggest the authors to implement this expansion by describing factors that affect the endothelial glycocalyx during aging and the effects of endothelial glycocalyx dysregulation as a result of aging on the integrity of the blood-brain barrier and the possible association of these events with the development of neurodegenerative diseases. To complete the topic, the effects of aging on the glycocalyx of immune cells that may affect the CNS should also be described.

The authors deal exclusively with Alzheimer's disease in section 4.4, where they describe glycosylation disorders and neurodegenerative diseases, i.e. diseases associated with aging. I suggest the authors also describe other age-related neurodegenerative diseases, e.g. Parkinson's disease, which are also associated with these changes.

The subtitles for sections 3 and 4 are the same. This should be corrected.

Name of Table 1 omitted

In some places in the text letters are missing in words, the use of whole words after the introduction of abbreviations, etc., which need to be corrected.

Author Response

Comment 1: Although the section of the article devoted to the effects of aging on galectins in the CNS is extensive, the section of this article devoted to the effects of aging on the glycocalyx is scarce and should be expanded. I suggest the authors to implement this expansion by describing factors that affect the endothelial glycocalyx during aging and the effects of endothelial glycocalyx dysregulation as a result of aging on the integrity of the blood-brain barrier and the possible association of these events with the development of neurodegenerative diseases. To complete the topic, the effects of aging on the glycocalyx of immune cells that may affect the CNS should also be described.

Response : We agree with your statements and sincerely thank you for the valuable suggestions. Circulatory problems in the brain are a major driving force behind aging-associated diseases. However, the endothelial glycocalyx is a more complex structure than merely a cell surface; it contains diverse proteoglycans, and alterations in this layer can lead to blood–brain barrier (BBB) disruption. This disruption promotes immune cell infiltration, resulting in increased inflammatory events. Furthermore, changes in the integrity of the immune cell surface glycocalyx affect their communication capacity and activation state, which in turn contributes to chronic intraparenchymal inflammation. We have expanded our review to address these topics in detail in lines 338 to 372 and 413-422.

Comment 2: The authors deal exclusively with Alzheimer's disease in section 4.4, where they describe glycosylation disorders and neurodegenerative diseases, i.e. diseases associated with aging. I suggest the authors also describe other age-related neurodegenerative diseases, e.g. Parkinson's disease, which are also associated with these changes.

Response: We agree with the reviewer’s criticism and sincerely thank you for your valuable remarks. It is indeed important to highlight the role of glycosylation in other aging-associated neurodegenerative diseases, such as Parkinson’s disease. Although Parkinson’s disease is primarily characterized by selective neuronal loss in the substantia nigra through a complex mechanism, modifications of the endothelial glycocalyx have also been observed. Moreover, changes in the glycation status of alpha-synuclein increase the toxicity of this key protein. We have incorporated this information into the manuscript in lines 355 to 360.

Comment 3:  The subtitles for sections 3 and 4 are the same. This should be corrected.

ResponseWe have corrected the errors and reorganized the subtitles accordingly.

Comment 4: Name of Table 1 omitted

Response: We have added the title for Table 1.

Comment 5: In some places in the text letters are missing in words, the use of whole words after the introduction of abbreviations, etc., which need to be corrected.

Response: We have reviewed the article and attempted to correct the mentioned mistakes. 

Reviewer 2 Report

Comments and Suggestions for Authors

The study titled ”Sweet Aging: Glycocalyx and Galectins in CNS Aging and Neurodegenerative” reviewed the research based on  the mechanisms of inflammation mediated by galectin related to glycocalyx during Aging and aging-related neurodegenerative disorders.

The review is well explained and easy to understand. However, in order to strengthen their ideas exposed in the manuscript they they should organize these into a prior index. Moreover, they should clearly explain specific points of table 1.

Main Remarks

  1. In table 1 the authors should explain why there aren't indicated ligands in some studies.
  2. In Figure 1 the authors should use the letter of the legend in a higher quality.
  3. Typo errors: In line 66 the authors should eliminate “.”  from “.Glycosylation”

In lines 104-105 the text appears discontinued between “and” and “another distinct..”

The authors should indicate the reference after text eliminating the “.” i.e. In line 112 “N glycans. [17].” And in line 382 “disease progression.[82].”

Author Response

Comment 1 : In table 1 the authors should explain why there aren't indicated ligands in some studies.

Response 1 : We agree with the comments, and our explanation is as follows: the cited articles do not identify the specific proteins that interact with galectins. Galectins can bind to other proteins either through carbohydrate-dependent mechanisms or via direct protein–protein interactions. In carbohydrate-dependent binding, galectins utilize their carbohydrate recognition domain (CRD) to couple with receptors. This type of interaction often requires higher concentrations of galectins to elicit measurable biological effects, as the mechanism involves lattice reorganization of cell surface proteins. This complexity makes it challenging to identify the exact interaction partners of galectins. The following article discusses and compares CRD-mediated and protein–protein interactions of galectin-1 in tumor cells:
Vas V, Fajka-Boja R, Ion G, Dudics V, Monostori E, Uher F. Biphasic effect of recombinant galectin-1 on the growth and death of early hematopoietic cells. Stem Cells. 2005 Feb;23(2):279–87. doi: 10.1634/stemcells.2004-0084. PMID: 15671150.

Comment 2: In Figure 1 the authors should use the letter of the legend in a higher quality.

Response: We have increased the resolution of the image to 600 dpi.  

Comments 3In lines 104-105 the text appears discontinued between “and” and “another distinct..”

The authors should indicate the reference after text eliminating the “.” i.e. In line 112 “N glycans. [17].” And in line 382 “disease progression.[82].”

Response: We have corrected the highlighted mistake.

Reviewer 3 Report

Comments and Suggestions for Authors

This manuscript addresses an important and timely topic, exploring the relationship between aging, glycosylation changes, lectin-mediated signaling, and neuroinflammation, with a particular focus on the role of galectins and glycocalyx modifications in the central nervous system (CNS). The work is well-organized and provides a comprehensive overview of current knowledge. While the review is overall well-constructed, I have a few suggestions for minor clarifications and refinements that could further enhance the clarity, precision, and impact of the manuscript.

  1. Clarification on glycocalyx localization (lines 58-60): The text mentions "ligands localized inside the membrane," but glycans are typically located on the extracellular surface as part of the glycocalyx. A slight rewording to clarify this point would strengthen accuracy.
  2. Definition of S-type lectins (lines 99-100): It may help to refine the definition, emphasizing that S-type lectins require a reducing environment for maintaining glycan-binding activity, rather than depending on disulfide bonds. Additionally, it is now recognized that many galectins discovered later do not possess the cysteine richness seen in galectin-1.
  3. Binding specificity for α2,6-sialylated glycans (lines 111-113): It would be helpful to clarify that modifications at the C6 position of galactose generally reduce galectin binding affinity. Although galectin-3 and galectin-9 can tolerate α2,6-sialylated structures to some extent, their overall binding strength remains decreased compared to α2,3-sialylated structures.

  4. Oxidized galectin-1 mechanisms (lines 140-141): It would be beneficial to mention that oxidized galectin-1 loses its glycan-binding ability but still promotes neurite outgrowth via alternative mechanisms.

  5. Clarification of Galectin-1 and CD45 interaction (lines 207-208): To improve precision, it could be helpful to specify that galectin-1 binds O-glycans on CD45, modulating CD45 activity.

  6. Correction on galectin in Huntington's disease (lines 221-222): It appears that galectin-3, rather than galectin-1, is involved in inflammasome activation in Huntington's disease. A minor correction would ensure factual consistency.

  7. Explanation of proliferation vs neurogenesis effects in SGZ (lines 280-281): The manuscript mentions that galectin-1 promotes neural progenitor proliferation while inhibiting neurogenesis, which may appear contradictory. Clarifying that galectin-1 increases proliferation of progenitor cells but may simultaneously inhibit their differentiation into mature neurons would resolve the inconsistency and better align with current understanding of hippocampal neurogenesis regulation.
  8. Link between Siglec signaling loss and galectin-3 upregulation (lines 364-368): Adding a sentence bridging these two events would help highlight the continuum of microglial activation processes.

  9. Minor typographical errors (e.g., "repaeat" in line 105 should be "repeat").

Author Response

Comment 1: Clarification on glycocalyx localization (lines 58-60): The text mentions "ligands localized inside the membrane," but glycans are typically located on the extracellular surface as part of the glycocalyx. A slight rewording to clarify this point would strengthen accuracy.

Response :We agree with the reviewer’s comment and have clarified the localization of the glycocalyx in the marked sentence on line 58.

Comment 2: Definition of S-type lectins (lines 99-100): It may help to refine the definition, emphasizing that S-type lectins require a reducing environment for maintaining glycan-binding activity, rather than depending on disulfide bonds. Additionally, it is now recognized that many galectins discovered later do not possess the cysteine richness seen in galectin-1.

Response: We agree with the reviewers and have clarified the definition of S-type lectins in line 100.

Comment 3: Binding specificity for α2,6-sialylated glycans (lines 111-113): It would be helpful to clarify that modifications at the C6 position of galactose generally reduce galectin binding affinity. Although galectin-3 and galectin-9 can tolerate α2,6-sialylated structures to some extent, their overall binding strength remains decreased compared to α2,3-sialylated structures.

Response: We completely agree and thank you for your assistance in clarifying the binding capacity of galectins to sialylated glycan structures. The text has been revised accordingly in lines 110–115.

Comment 4: Oxidized galectin-1 mechanisms (lines 140-141): It would be beneficial to mention that oxidized galectin-1 loses its glycan-binding ability but still promotes neurite outgrowth via alternative mechanisms.

Response: We completely agree with the reviewer’s comment. We have emphasized in the text (lines 141–143) that the neurite outgrowth–promoting capacity of galectin-1 is independent of its carbohydrate recognition.

Comment 5: Clarification of Galectin-1 and CD45 interaction (lines 207-208): To improve precision, it could be helpful to specify that galectin-1 binds O-glycans on CD45, modulating CD45 activity.

Response : We agree with the reviewer and have clarified in the text (lines 208–209) that there is a direct interaction between galectin-1 and core2-O glycans on CD45 during microglial reprogramming.

Comment 6:  It appears that galectin-3, rather than galectin-1, is involved in inflammasome activation in Huntington's disease. A minor correction would ensure factual consistency.

Response 6We agree with the reviewer’s comment and have corrected the mistake in the text at line 222

Comment 7Explanation of proliferation vs neurogenesis effects in SGZ (lines 280-281): The manuscript mentions that galectin-1 promotes neural progenitor proliferation while inhibiting neurogenesis, which may appear contradictory. Clarifying that galectin-1 increases proliferation of progenitor cells but may simultaneously inhibit their differentiation into mature neurons would resolve the inconsistency and better align with current understanding of hippocampal neurogenesis regulation.

Response:We completely agree with the comments and have clarified the text as requested by the reviewers in lines 278-281.

Comment 8: Link between Siglec signaling loss and galectin-3 upregulation (lines 364-368): Adding a sentence bridging these two events would help highlight the continuum of microglial activation processes.

Response 8: We agree with the comment and have emphasized in the text (lines 391–393) that activated microglia lacking ITIM signaling also release galectin-3, which further activates them through TLR4 and TREM2 receptors.

Comment 9: Minor typographical errors (e.g., "repaeat" in line 105 should be "repeat").

Response: We have made efforts to avoid and correct any typographical errors, including using the MS Word spell checker for assistance.

Round 2

Reviewer 1 Report

Comments and Suggestions for Authors

The authors have successfully made changes to the manuscript in line with the suggestions given.